# Efficient Use of Limited-Memory Accelerators for Linear Learning on Heterogeneous Systems

**Celestine Dünner**
IBM Research - Zurich
Switzerland
cdu@zurich.ibm.com

**Thomas Parnell**
IBM Research - Zurich
Switzerland
tpa@zurich.ibm.com

**Martin Jaggi**
EPFL
Switzerland
martin.jaggi@epfl.ch

## Abstract

We propose a generic algorithmic building block to accelerate training of machine learning models on heterogeneous compute systems. Our scheme allows to efficiently employ compute accelerators such as GPUs and FPGAs for the training of large-scale machine learning models, when the training data exceeds their memory capacity. Also, it provides adaptivity to any system's memory hierarchy in terms of size and processing speed. Our technique is built upon novel theoretical insights regarding primal-dual coordinate methods, and uses duality gap information to dynamically decide which part of the data should be made available for fast processing. To illustrate the power of our approach we demonstrate its performance for training of generalized linear models on a large-scale dataset exceeding the memory size of a modern GPU, showing an order-of-magnitude speedup over existing approaches.

## 1 Introduction

As modern compute systems rapidly increase in size, complexity and computational power, they become less homogeneous. Today's systems exhibit strong heterogeneity at many levels: in terms of compute parallelism, memory size and access bandwidth, as well as communication bandwidth between compute nodes (e.g., computers, mobile phones, server racks, GPUs, FPGAs, storage nodes etc.). This increasing heterogeneity of compute environments is posing new challenges for the development of efficient distributed algorithms. That is to optimally exploit individual compute resources with very diverse characteristics without suffering from the I/O cost of exchanging data between them.

In this paper, we focus on the task of training large scale machine learning models in such heterogeneous compute environments and propose a new generic algorithmic building block to efficiently distribute the workload between heterogeneous compute units. Assume two compute units, denoted $\mathcal{A}$ and $\mathcal{B}$, which differ in compute power as well as memory capacity as illustrated in Figure 1. The computational power of unit $\mathcal{A}$ is smaller and its memory capacity is larger relative to its peer unit $\mathcal{B}$ (i.e., we assume that the training data fits into the memory of $\mathcal{A}$, but not into $\mathcal{B}$'s). Hence, on the computationally more powerful unit $\mathcal{B}$, only part of the data can be processed at any given time. The two units, $\mathcal{A}$ and $\mathcal{B}$, are able to communicate with each other over some interface, however there is cost associated with doing so.

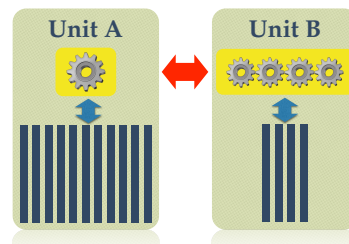

Figure 1: Compute units $\mathcal{A}$, $\mathcal{B}$ with different memory size, bandwidth and compute power.

This generic setup covers many essential elements of modern machine learning systems. A typical example is that of accelerator units, such as a GPUs or FPGAs, augmenting traditional computers

or servers. While such devices can offer a significant increase in computational power due to their massively parallel architectures, their memory capacity is typically very limited. Another example can be found in hierarchical memory systems where data in the higher level memory can be accessed and hence processed faster than data in the – typically larger – lower level memory. Such memory systems are spanning from fast on-chip caches on one extreme to slower hard drives on the other extreme.

The core question we address in this paper is the following: *How can we efficiently distribute the workload between heterogeneous units $\mathcal{A}$ and $\mathcal{B}$ in order to accelerate large scale learning?*

The generic algorithmic building block we propose systematically splits the overall problem into two workloads, a more data-intensive but less compute-intensive part for unit $\mathcal{A}$ and a more compute-intensive but less data-intensive part for $\mathcal{B}$. These workloads are then executed in parallel, enabling full utilization of both resources while keeping the amount of necessary communication between the two units minimal. Such a generic algorithmic building block is useful much more widely than just for training on two heterogeneous compute units – it can serve as a component of larger training algorithms or pipelines thereof. In a distributed training setting, our scheme allows each individual node to locally benefit from its own accelerator, therefore speeding up the overall task on a cluster, e.g., as part of [14] or another distributed algorithm. Orthogonal to such a horizontal application, our scheme can also be used as a building block vertically integrated in a system, serving the efficiency of several levels of the memory hierarchy of a given compute node.

**Related Work.**   The most popular existing approach to deal with memory limitations is to process data in batches. For example, for the special case of SVMs, [16] splits data samples into blocks which are then loaded and processed sequentially (on $\mathcal{B}$), in the setting of limited RAM and the full data residing on disk. This approach enables contiguous chunks of data to be loaded which is beneficial in terms of I/O overhead; it however treats samples uniformly. The same holds for [15] where blocks to be loaded are selected randomly. Later, in [2, 7] it is proposed to selectively load and keep informative samples in memory in order to reduce disk access, but this approach is specific to support vectors and is unable to theoretically quantify the possible speedup.

In this work, we propose a novel, theoretically-justified scheme to efficiently deal with memory limitations in the heterogeneous two-unit setting illustrated in Figure 1. Our scheme can be applied to a broad class of machine learning problems, including generalized linear models, empirical risk minimization problems with a strongly convex regularizer, such as SVM, as well as sparse models, such as Lasso. In contrast to the related line of research [16, 2, 7], our scheme is designed to take full advantage of both compute resources $\mathcal{A}$ and $\mathcal{B}$ for training, by systematically splitting the workload among $\mathcal{A}$ and $\mathcal{B}$ in order to adapt to their specific properties and to the available bandwidth between them. At the heart of our approach lies a smart data selection scheme using coordinate-wise duality gaps as selection criteria. Our theory will show that our selection scheme provably improves the convergence rate of training overall, by explicitly quantifying the benefit over uniform sampling. In contrast, existing work [2, 7] only showed that the linear convergence rate on SVMs is preserved asymptotically, but not necessarily improved.

A different line of related research is steepest coordinate selection. It is known that steepest coordinate descent can converge much faster than uniform [8] for single coordinate updates on smooth objectives, however it typically does not perform well for general convex problems, such as those with $L1$ regularization. In our work, we overcome this issue by using the generalized primal-dual gaps [4] which do extend to $L1$ problems. Related to this notion, [3, 9, 11] have explored the use of similar information as an adaptive measure of importance, in order to adapt the sampling probabilities of coordinate descent. Both this line of research as well as steepest coordinate descent [8] are still limited to single coordinate updates, and cannot be readily extended to arbitrary accuracy updates on a larger subset of coordinates (performed per communication round) as required in our heterogeneous setting.

**Contributions.**   The main contributions of this work are summarized as follows:

- We analyze the per-iteration-improvement of primal-dual block coordinate descent and how it depends on the selection of the active coordinate block at that iteration. Further, we extend the convergence theory to arbitrary approximate updates on the coordinate subsets. We propose a novel dynamic selection scheme for blocks of coordinates, which relies on coordinate-wise duality gaps, and precisely quantify the speedup of the convergence rate over uniform sampling.

- Our theoretical findings result in a scheme for learning in heterogeneous compute environments which is easy to use, theoretically justified and versatile in that it can be adapted to given resource constraints, such as memory, computation and communication. Furthermore our scheme enables parallel execution between, and also within, two heterogeneous compute units.

- For the example of joint training in a CPU plus GPU environment – which is very challenging for data-intensive work loads – we demonstrate a more than $10\times$ speed-up over existing methods for limited-memory training.

## 2 Learning Problem

For the scope of this work we focus on the training of convex generalized linear models of the form

$$\min_{\boldsymbol{\alpha}\in\mathbb{R}^n} \quad \mathcal{O}(\boldsymbol{\alpha}) := f(A\boldsymbol{\alpha}) \, + \, g(\boldsymbol{\alpha}) \tag{1}$$

where $f$ is a smooth function and $g(\boldsymbol{\alpha}) = \sum_i g_i(\alpha_i)$ is separable, $\boldsymbol{\alpha} \in \mathbb{R}^n$ describes the parameter vector and $A = [\mathbf{a}_1, \mathbf{a}_2, \ldots, \mathbf{a}_n] \in \mathbb{R}^{d\times n}$ the data matrix with column vectors $\mathbf{a}_i \in \mathbb{R}^d$. This setting covers many prominent machine learning problems, including generalized linear models as used for regression, classification and feature selection. To avoid confusion, it is important to distinguish the two main application classes: On one hand, we cover empirical risk minimization (ERM) problems with a strongly convex regularizer such as $L_2$-regularized SVM – where $\boldsymbol{\alpha}$ then is the dual variable vector and $f$ is the smooth regularizer conjugate, as in SDCA [13]. On the other hand, we also cover the class of sparse models such as Lasso or ERM with a sparse regularizer – where $f$ is the data-fit term and $g$ takes the role of the non-smooth regularizer, so $\boldsymbol{\alpha}$ are the original primal parameters.

**Duality Gap.** Through the perspective of Fenchel-Rockafellar duality, one can, for any primal-dual solution pair $(\boldsymbol{\alpha}, \mathbf{w})$, define the non-negative duality gap for (1) as

$$\text{gap}(\boldsymbol{\alpha}; \mathbf{w}) \quad := \quad f(A\boldsymbol{\alpha}) + g(\boldsymbol{\alpha}) + f^*(\mathbf{w}) + g^*(-A^\top\mathbf{w}) \tag{2}$$

where the functions $f^*$, $g^*$ in (2) are defined as the *convex conjugate*[1] of their corresponding counterparts $f, g$ [1]. Let us consider parameters $\mathbf{w}$ that are optimal relative to a given $\boldsymbol{\alpha}$, i.e.,

$$\mathbf{w} := \mathbf{w}(\boldsymbol{\alpha}) = \nabla f(A\boldsymbol{\alpha}), \tag{3}$$

which implies $f(A\boldsymbol{\alpha}) + f^*(\mathbf{w}) = \langle A\boldsymbol{\alpha}, \mathbf{w}\rangle$. In this special case, the duality gap (2) simplifies and becomes separable over the columns $\mathbf{a}_i$ of $A$ and the corresponding parameter weights $\alpha_i$ given $\mathbf{w}$. We will later exploit this property to quantify the suboptimality of individual coordinates.

$$\text{gap}(\boldsymbol{\alpha}) = \sum_{i\in[n]} \text{gap}_i(\alpha_i), \quad \text{where} \quad \text{gap}_i(\alpha_i) := \mathbf{w}^\top\mathbf{a}_i\alpha_i + g_i(\alpha_i) + g_i^*(-\mathbf{a}_i^\top\mathbf{w}). \tag{4}$$

**Notation.** For the remainder of the paper we use $\mathbf{v}_{[\mathcal{P}]}$ to denote a vector $\mathbf{v}$ with non-zero entries only for the coordinates $i \in \mathcal{P} \subseteq [n] = \{1, \ldots, n\}$. Similarly we write $A_{[\mathcal{P}]}$ to denote the matrix $A$ composing only of columns indexed by $i \in \mathcal{P}$.

## 3 Approximate Block Coordinate Descent

The theory we present in this section serves to derive a theoretical framework for our heterogeneous learning scheme presented in Section 4. Therefore, let us consider the generic block minimization scheme described in Algorithm 1 to train generalized linear models of the form (1).

### 3.1 Algorithm Description

In every round $t$, of Algorithm 1, a block $\mathcal{P}$ of $m$ coordinates of $\boldsymbol{\alpha}$ is selected according to an arbitrary selection rule. Then, an update is computed on this block of coordinates by optimizing

$$\underset{\Delta\boldsymbol{\alpha}_{[\mathcal{P}]}\in\mathbb{R}^n}{\arg\min} \quad \mathcal{O}(\boldsymbol{\alpha} + \Delta\boldsymbol{\alpha}_{[\mathcal{P}]}) \tag{5}$$

where an arbitrary solver can be used to find this update. This update is not necessarily perfectly optimal but of a relative accuracy $\theta$, in the following sense of approximation quality:

| **Algorithm 1** Approximate Block CD | **Algorithm 2** DuHL |
|---|---|
| 1: Initialize $\boldsymbol{\alpha}^{(0)} := \mathbf{0}$ | 1: Initialize $\boldsymbol{\alpha}^{(0)} := \mathbf{0}$, $\mathbf{z} := \mathbf{0}$ |
| 2: **for** $t = 0, 1, 2, ...$ **do** | 2: **for** $t = 0, 1, 2, ...$ |
| 3:     select a subset $\mathcal{P}$ with $|\mathcal{P}| = m$ | 3:     determine $\mathcal{P}$ according to (13) |
| 4:     $\Delta\boldsymbol{\alpha}_{[\mathcal{P}]} \leftarrow \theta$-approx. solution to (5) | 4:     refresh memory $\mathcal{B}$ to contain $A_{[\mathcal{P}]}$. |
| 5:     $\boldsymbol{\alpha}^{(t+1)} := \boldsymbol{\alpha}^{(t)} + \Delta\boldsymbol{\alpha}_{[\mathcal{P}]}$ | 5:     **on $\mathcal{B}$ do:** |
| 6: **end for** | 6:         $\Delta\boldsymbol{\alpha}_{[\mathcal{P}]} \leftarrow \theta$-approx. solution to (12) |
| | 7:     **in parallel on $\mathcal{A}$ do:** |
| | 8:         **while** $\mathcal{B}$ not finished |
| | 9:             sample $j \in [n]$ |
| | 10:           update $z_j := \mathrm{gap}_j(\alpha_j^{(t)})$ |
| | 11:     $\boldsymbol{\alpha}^{(t+1)} := \boldsymbol{\alpha}^{(t)} + \Delta\boldsymbol{\alpha}_{[\mathcal{P}]}$ |

**Definition 1** ($\theta$-Approximate Update). The block update $\Delta\boldsymbol{\alpha}_{[\mathcal{P}]}$ is *$\theta$-approximate* iff

$$\exists\theta \in [0,1]: \quad \mathcal{O}(\boldsymbol{\alpha} + \Delta\boldsymbol{\alpha}_{[\mathcal{P}]}) \leq \theta\mathcal{O}(\boldsymbol{\alpha} + \Delta\boldsymbol{\alpha}^{\star}_{[\mathcal{P}]}) + (1-\theta)\mathcal{O}(\boldsymbol{\alpha}) \tag{6}$$

where $\Delta\boldsymbol{\alpha}^{\star}_{[\mathcal{P}]} \in \arg\min_{\Delta\boldsymbol{\alpha}_{[\mathcal{P}]}\in\mathbb{R}^n} \mathcal{O}(\boldsymbol{\alpha} + \Delta\boldsymbol{\alpha}_{[\mathcal{P}]})$.

### 3.2 Convergence Analysis

In order to derive a precise convergence rate for Algorithm 1 we build on the convergence analysis of [4, 13]. We extend their analysis of stochastic coordinate descent in two ways: 1) to a block coordinate scheme with approximate coordinate updates, and 2) to explicitly cover the importance of each selected coordinate, as opposed to uniform sampling.

We define

$$\rho_{t,\mathcal{P}} := \frac{\frac{1}{m}\sum_{j\in\mathcal{P}}\mathrm{gap}_j(\alpha_j^{(t)})}{\frac{1}{n}\sum_{j\in[n]}\mathrm{gap}_j(\alpha_j^{(t)})} \tag{7}$$

which quantifies how much the coordinates $i \in \mathcal{P}$ of $\boldsymbol{\alpha}^{(t)}$ contribute to the global duality gap (2). Thus giving a measure of suboptimality for these coordinates. In Algorithm 1 an arbitrary selection scheme (deterministic or randomized) can be applied and our theory will explain how the convergence of Algorithm 1 depends on the selection through the distribution of $\rho_{t,\mathcal{P}}$. That is, for strongly convex functions $g_i$, we found that the per-step improvement in suboptimality is proportional to $\rho_{t,\mathcal{P}}$ of the specific coordinate block $\mathcal{P}$ being selected at that iteration $t$:

$$\epsilon^{(t+1)} \leq (1 - \rho_{t,\mathcal{P}}\theta c)\,\epsilon^{(t)} \tag{8}$$

where $\epsilon^{(t)} := \mathcal{O}(\boldsymbol{\alpha}^{(t)}) - \mathcal{O}(\boldsymbol{\alpha}^{\star})$ measures the suboptimality of $\boldsymbol{\alpha}^{(t)}$ and $c > 0$ is a constant which will be specified in the following theorem. A similar dependency on $\rho_{t,\mathcal{P}}$ can also be shown for non-strongly convex functions $g_i$, leading to our two main convergence results for Algorithm 1:

**Theorem 1.** *For Algorithm 1 running on* (1) *where $f$ is $L$-smooth and $g_i$ is $\mu$-strongly convex with $\mu > 0$ for all $i \in [n]$, it holds that*

$$\mathbb{E}_{\mathcal{P}}[\epsilon^{(t)} \,|\, \boldsymbol{\alpha}^{(0)}] \leq \left(1 - \eta_{\mathcal{P}}\frac{m}{n}\frac{\mu}{\sigma L + \mu}\right)^t \epsilon^{(0)} \tag{9}$$

*where $\sigma := \|A_{[\mathcal{P}]}\|_{op}^2$ and $\eta_{\mathcal{P}} := \min_t \theta\,\mathbb{E}_{\mathcal{P}}[\rho_{t,\mathcal{P}} \,|\, \boldsymbol{\alpha}^{(t)}]$. Expectations are over the choice of $\mathcal{P}$.*

That is, for strongly convex $g_i$, Algorithm 1 has a linear convergence rate. This was shown before in [13, 4] for the special case of exact coordinate updates. In strong contrast to earlier coordinate descent analyses which build on random uniform sampling, our theory explicitly quantifies the impact of the sampling scheme on the convergence through $\rho_{t,\mathcal{P}}$. This allows one to benefit from smart selection and provably improve the convergence rate by taking advantage of the inhomogeneity of the duality gaps. The same holds for non-strongly convex functions $g_i$:

**Theorem 2.** *For Algorithm 1 running on* (1) *where $f$ is $L$-smooth and $g_i$ has $B$-bounded support for all $i \in [n]$, it holds that*

$$\mathbb{E}_{\mathcal{P}}[\epsilon^{(t)} \,|\, \boldsymbol{\alpha}^{(0)}] \leq \frac{1}{\eta_{\mathcal{P}} m} \frac{2\gamma n^2}{2n + t - t_0} \tag{10}$$

*with $\gamma := 2LB^2\sigma$ where $\sigma := \|A_{[\mathcal{P}]}\|_{op}^2$ and $t \geq t_0 = \max\left\{0, \frac{n}{m}\log\left(\frac{2\eta m\epsilon^{(0)}}{n\gamma}\right)\right\}$ where $\eta_{\mathcal{P}} := \min_t \theta\, \mathbb{E}_{\mathcal{P}}[\rho_{t,\mathcal{P}} \,|\, \boldsymbol{\alpha}^{(t)}]$. Expectations are over the choice of $\mathcal{P}$.*

**Remark 1.** *Note that for uniform selection, our proven convergence rates for Algorithm 1 recover classical primal-dual coordinate descent [4, 13] as a special case, where in every iteration a single coordinate is selected and each update is solved exactly, i.e., $\theta = 1$. In this case $\rho_{t,\mathcal{P}}$ measures the contribution of a single coordinate to the duality gap. For uniform sampling, $\mathbb{E}_{\mathcal{P}}[\rho_{t,\mathcal{P}} \,|\, \boldsymbol{\alpha}^{(t)}] = 1$ and hence $\eta_{\mathcal{P}} = 1$ which recovers [4, Theorems 8 and 9].*

### 3.3 Gap-Selection Scheme

The convergence results of Theorems 1 and 2 suggest that the optimal rule for selecting the block of coordinates $\mathcal{P}$ in step 3 of Algorithm 1, leading to the largest improvement in that step, is the following:

$$\mathcal{P} := \underset{\mathcal{P} \subset [n]: |\mathcal{P}| = m}{\arg\max} \sum_{j \in \mathcal{P}} \mathrm{gap}_j\left(\alpha_j^{(t)}\right). \tag{11}$$

This scheme maximizes $\rho_{t,\mathcal{P}}$ at every iterate. Furthermore, the selection scheme (11) guarantees $\rho_{t,\mathcal{P}} \geq 1$ which quantifies the relative gain over random uniform sampling. In contrast to existing importance sampling schemes [17, 12, 5] which assign static probabilities to individual coordinates, our selection scheme (11) is dynamic and adapts to the current state $\boldsymbol{\alpha}^{(t)}$ of the algorithm, similar to that used in [9, 11] in the standard non-heterogeneous setting.

## 4 Heterogeneous Training

In this section we build on the theoretical insight of the previous section to tackle the main objective of this work: How can we efficiently distribute the workload between two heterogeneous compute units $\mathcal{A}$ and $\mathcal{B}$ to train a large-scale machine learning problem where $\mathcal{A}$ and $\mathcal{B}$ fulfill the following two assumptions:

**Assumption 1** (Difference in Memory Capacity). *Compute unit $\mathcal{A}$ can fit the whole dataset in its memory and compute unit $\mathcal{B}$ can only fit a subset of the data. Hence, $\mathcal{B}$ only has access to $A_{[\mathcal{P}]}$, a subset $\mathcal{P}$ of $m$ columns of $A$, where $m$ is determined by the memory size of $\mathcal{B}$.*

**Assumption 2** (Difference in Computational Power). *Compute unit $\mathcal{B}$ can access and process data faster than compute unit $\mathcal{A}$.*

### 4.1 DUHL: A Duality Gap-Based Heterogeneous Learning Scheme

We propose a duality gap-based heterogeneous learning scheme, henceforth referring to as DUHL, for short. DUHL is designed for efficient training on heterogeneous compute resources as described above. The core idea of DUHL is to identify a block $\mathcal{P}$ of coordinates which are most relevant to improving the model at the current stage of the algorithm, and have the corresponding data columns, $A_{[\mathcal{P}]}$, residing locally in the memory of $\mathcal{B}$. Compute unit $\mathcal{B}$ can then exploit its superior compute power by using an appropriate solver to locally find a block coordinate update $\Delta\boldsymbol{\alpha}_{[\mathcal{P}]}$. At the same time, compute unit $\mathcal{A}$, is assigned the task of updating the block $\mathcal{P}$ of important coordinates as the algorithm proceeds and the iterates change. Through this split of workloads DUHL enables full utilization of both compute units $\mathcal{A}$ and $\mathcal{B}$. Our scheme, summarized in Algorithm 2, fits the theoretical framework established in the previous section and can be viewed as an instance of Algorithm 1, implementing a time-delayed version of the duality gap-based selection scheme (11).

**Local Subproblem.** In the heterogeneous setting compute unit $\mathcal{B}$ only has access to its local data $A_{[\mathcal{P}]}$ and some current state $\mathbf{v} := A\boldsymbol{\alpha} \in \mathbb{R}^d$ in order to compute a block update $\Delta\boldsymbol{\alpha}_{[\mathcal{P}]}$ in Step 4 of Algorithm 1. While for quadratic functions $f$ this information is sufficient to optimize (5), for non-quadratic functions $f$ we consider the following modified local optimization problem instead:

$$\underset{\Delta\boldsymbol{\alpha}_{[\mathcal{P}]} \in \mathbb{R}^n}{\arg\min}\ f(\mathbf{v}) + \langle \nabla f(\mathbf{v}), A\Delta\boldsymbol{\alpha}_{[\mathcal{P}]}\rangle + \frac{L}{2}\|A\Delta\boldsymbol{\alpha}_{[\mathcal{P}]}\|_2^2 + \sum_{i \in \mathcal{P}} g_i((\boldsymbol{\alpha} + \Delta\boldsymbol{\alpha}_{[\mathcal{P}]})_i). \tag{12}$$

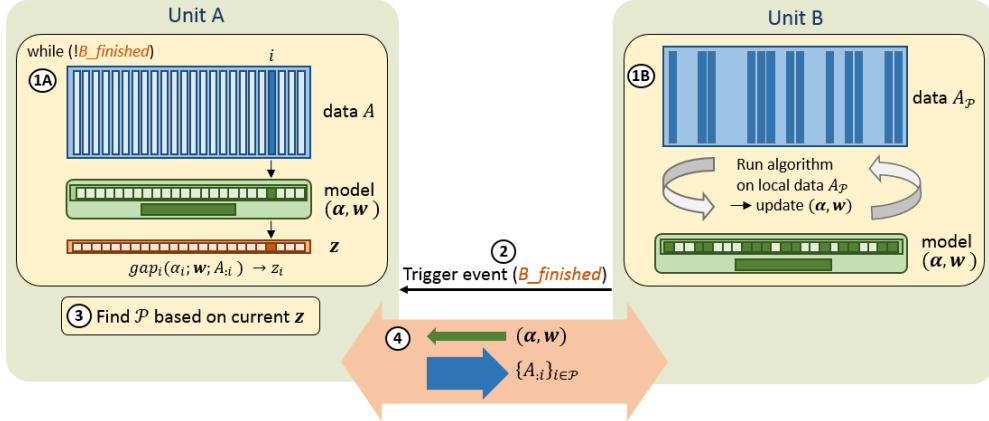

Figure 2: Illustration of one round of DUHL as described in Algorithm 2.

It can be shown that the convergence guarantees of Theorems 1 and 2 similarly hold if the block coordinate update in Step 4 of Algorithm 1 is computed on (12) instead of (5) (see Appendix C for more details).

**A Time-Delayed Gap Measure.** Motivated by our theoretical findings, we use the duality gap as a measure of importance for selecting which coordinates unit $\mathcal{B}$ is working on. However, a scheme as suggested in (11) is not suitable for our purpose since it requires knowledge of the duality gaps (4) for every coordinate $i$ at a given iterate $\boldsymbol{\alpha}^{(t)}$. For our scheme this would imply a computationally expensive selection step at the beginning of every round which has to be performed in sequence to the update step. To overcome this and enable parallel execution of the two workloads on $\mathcal{A}$ and $\mathcal{B}$, we propose to introduce a *gap memory*. This is an $n$-dimensional vector $\mathbf{z}$ where $z_i$ measures the importance of coordinate $\alpha_i$. We have $z_i := \mathrm{gap}(\alpha_i^{(t')})$ where $t' \in [0, t]$ and the different elements of $\mathbf{z}$ are allowed to be based on different, possibly stale iterates $\boldsymbol{\alpha}^{(t')}$. Thus, the entries of $\mathbf{z}$ can be continuously updated during the course of the algorithm. Then, at the beginning of every round the new block $\mathcal{P}$ is chosen based on the current state of $\mathbf{z}$ as follows:

$$\mathcal{P} := \underset{\mathcal{P} \subset [n] : |\mathcal{P}| = m}{\arg\max} \sum_{j \in \mathcal{P}} z_j. \tag{13}$$

In DUHL, keeping $\mathbf{z}$ up to date is the job of compute unit $\mathcal{A}$. Hence, while $\mathcal{B}$ is computing a block coordinate update $\Delta\boldsymbol{\alpha}_{[\mathcal{P}]}$, $\mathcal{A}$ updates $\mathbf{z}$ by randomly sampling from the entire training data. Then, as soon as $\mathcal{B}$ is done, the current state of $\mathbf{z}$ is used to determine $\mathcal{P}$ for the next round and data columns on $\mathcal{B}$ are replaced if necessary. The parallel execution of the two workloads during a single round of DUHL is illustrated in Figure 2. Note, that the freshness of the gap-memory $\mathbf{z}$ depends on the relative compute power of $\mathcal{A}$ versus $\mathcal{B}$, as well as $\theta$ which controls the amount of time spent computing on unit $\mathcal{B}$ in every round.

In Section 5.2 we will experimentally investigate the effect of staleness of the values $z_i$ on the convergence behavior of our scheme.

## 5 Experimental Results

For our experiments we have implemented DUHL for the particular use-case where $\mathcal{A}$ corresponds to a CPU with attached RAM and $\mathcal{B}$ corresponds to a GPU – $\mathcal{A}$ and $\mathcal{B}$ communicate over the PCIe bus. We use an 8-core Intel Xeon E5 x86 CPU with 64GB of RAM which is connected over PCIe Gen3 to an NVIDIA Quadro M4000 GPU which has 8GB of RAM. GPUs have recently experience a widespread adoption in machine learning systems and thus this hardware scenario is timely and highly relevant. In such a setting we wish to apply DUHL to efficiently populate the GPU memory and thereby making this part of the data available for fast processing.

**GPU solver.** In order to benefit from the enormous parallelism offered by GPUs and fulfill Assumption 2, we need a local solver capable of exploiting the power of the GPU. Therefore, we have chosen to implement the twice parallel, asynchronous version of stochastic coordinate descent

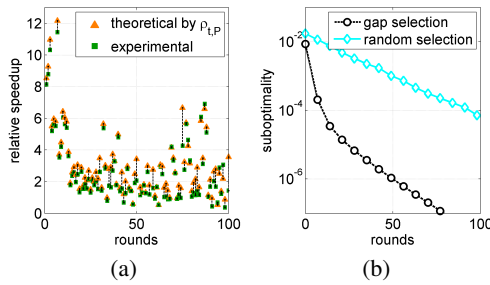
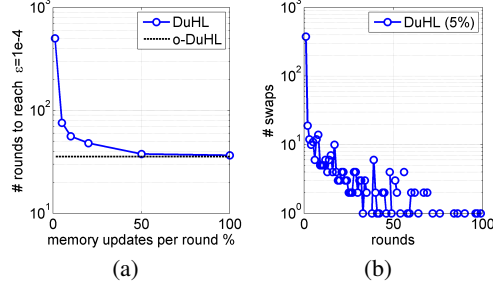

Figure 3: Validation of faster convergence: (a) theoretical quantity $\rho_{t,\mathcal{P}}$ (orange), versus the practically observed speedup (green) – both relative to the random scheme baseline, (b) convergence of gap selection compared to random selection.

Figure 4: Effect of stale entries in the gap memory of DUHL: (a) number of rounds needed to reach suboptimality $10^{-4}$ for different update frequencies compared to o-DUHL, (b) the number of data columns that are replaced per round for update frequency of $5\%$.

(TPA-SCD) that has been proposed in [10] for solving ridge regression. In this work we have generalized the implementation further so that it can be applied in a similar manner to solve the Lasso, as well as the SVM problem. For more details about the algorithm and how to generalize it we refer the reader to Appendix D.

## 5.1 Algorithm Behavior

Firstly, we will use the publicly available epsilon dataset from the LIBSVM website (a fully dense dataset with 400'000 samples and 2'000 features) to study the convergence behavior of our scheme. For the experiments in this section we assume that the GPU fits $25\%$ of the training data, i.e., $m = \frac{n}{4}$ and show results for training the sparse Lasso as well as the ridge regression model. For the Lasso case we have chosen the regularizer to obtain a support size of $\sim 12\%$ and we apply the coordinate-wise Lipschitzing trick [4] to the $L_1$-regularizer in order to allow the computation of the duality gaps. For computational details we refer the reader to Appendix E.

**Validation of Faster Convergence.** From our theory in Section 3.2 we expect that during any given round $t$ of Algorithm 1, the relative gain in convergence rate of one sampling scheme over the other should be quantified by the ratio of the corresponding values of $\eta_{t,\mathcal{P}} := \theta\rho_{t,\mathcal{P}}$ (for the respective block of coordinates processed in this round). To verify this, we trained a ridge regression model on the epsilon dataset implementing a) the gap-based selection scheme, (11), and b) random selection, fixing $\theta$ for both schemes. Then, in every round $t$ of our experiment, we record the value of $\rho_{t,\mathcal{P}}$ as defined in (7) and measure the relative gain in convergence rate of the gap-based scheme over the random scheme. In Figure 3(a) we plot the effective speedup of our scheme, and observe that this speedup almost perfectly matches the improvement predicted by our theory as measured by $\rho_{t,\mathcal{P}}$ - we observe an average deviation of $0.42$. Both speedup numbers are calculated relative to plain random selection. In Figure 3(b) we see that the gap-based selection can achieve a remarkable $10\times$ improvement in convergence over the random reference scheme. When running on sparse problems instead of ridge regression, we have observed $\rho_{t,\mathcal{P}}$ of the oracle scheme converging to $\frac{n}{m}$ within only a few iterations if the support of the problem is smaller than $m$ and fits on the GPU.

**Effect of Gap-Approximation.** In this section we study the effect of using stale, inconsistent gap-memory entries for selection on the convergence of DUHL. While the freshness of the memory entries is, in reality, determined by the relative compute power of unit $\mathcal{B}$ over unit $\mathcal{A}$ and the relative accuracy $\theta$, in this experiment we artificially vary the number of gap updates performed during each round while keeping $\theta$ fixed. We train the Lasso model and show, in Figure 4(a), the number of rounds needed to reach a suboptimality of $10^{-4}$, as a function of the number of gap entries updated per round. As a reference we show o-DUHL which has access to an oracle providing the true duality gaps. We observe that our scheme is quite robust to stale gap values and can achieve performance within a factor of two over the oracle scheme up to an average delay of 20 iterations. As the update frequency decreases we observed that the convergence slows down in the initial rounds because the algorithm needs more rounds until the active set of the sparse problem is correctly detected.

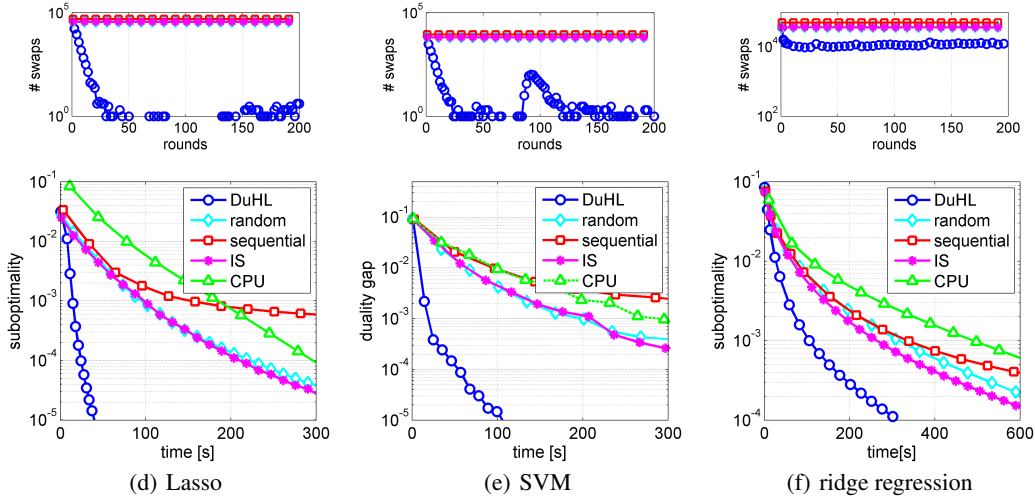

(d) Lasso        (e) SVM        (f) ridge regression

Figure 5: Performance results of DUHL on the 30GB ImageNet dataset. I/O cost (top) and convergence behavior (bottom) for Lasso, SVM and ridge regression.

**Reduced I/O operations.** The efficiency of our scheme regarding I/O operations is demonstrated in Figure 4(b), where we plot the number of data columns that are replaced on $\mathcal{B}$ in every round of Algorithm 2. Here the Lasso model is trained assuming a gap update frequency of $5\%$. We observe that the number of required I/O operations of our scheme is decreasing over the course of the algorithm. When increasing the freshness of the gap memory entries we could see the number of swaps go to zero faster.

## 5.2 Reference Schemes

In the following we compare the performance of our scheme against four reference schemes. We compare against the most widely-used scheme for using a GPU to accelerate training when the data does not fit into the memory of the GPU, that is the *sequential block selection* scheme presented in [16]. Here the data columns are split into blocks of size $m$ which are sequentially put on the GPU and operated on (the data is efficiently copied to the GPU as a contiguous memory block).

We also compare against importance sampling as presented in [17], which we refer to as IS. Since probabilities assigned to individual data columns are static we cannot use them as importance measures in a deterministic selection scheme. Therefore, in order to apply importance sampling in the heterogeneous setting, we non-uniformly sample $m$ data-columns to reside inside the GPU memory in every round of Algorithm 2 and have the CPU determine the new set in parallel. As we will see, data column norms often come with only small variance, in particular for dense datasets. Therefore, importance sampling often fails to give a significant gain over uniformly random selection.

Additionally, we compare against a single-threaded CPU implementation of a stochastic coordinate descent solver to demonstrate that with our scheme, the use of a GPU in such a setting indeed yields a significant speedup over a basic CPU implementation despite the high I/O cost of repeatedly copying data on and off the GPU memory. To the best of our knowledge, we are the first to demonstrate this.

For all competing schemes, we use TPA-SCD as the solver to efficiently compute the block update $\Delta\boldsymbol{\alpha}_{[\mathcal{P}]}$ on the GPU. The accuracy $\theta$ of the block update computed in every round is controlled by the number of randomized passes of TPA-SCD through the coordinates of the selected block $\mathcal{P}$. For a fair comparison we optimize this parameter for the individual schemes.

## 5.3 Performance Analysis of DUHL

For our large-scale experiments we use an extended version of the Kaggle Dogs vs. Cats ImageNet dataset as presented in [6], where we additionally double the number of samples, while using single precision floating point numbers. The resulting dataset is fully dense and consists of 40'000 samples and 200'704 features, resulting in over 8 billion non-zero elements and a data size of 30GB. Since the memory capacity of our GPU is 8GB, we can put $\sim 25\%$ of the data on the GPU. We will show

results for training a sparse Lasso model, ridge regression as well as linear $L_2$-regularized SVM. For Lasso we choose the regularization to achieve a support size of $12\%$, whereas for SVM the regularizer was chosen through cross-validation. For all three tasks, we compare the performance of DUHL to sequential block selection, random selection, selection through importance sampling (IS) all on GPU, as well as a single-threaded CPU implementation. In Figure 5(d) and 5(e) we demonstrate that for Lasso as well as SVM, DUHL converges $10\times$ faster than any reference scheme. This gain is achieved by improved convergence – quantified through $\rho_{t,\mathcal{P}}$ – as well as through reduced I/O cost, as illustrated in the top plots of Figure 5, which show the number of data columns replaced per round. The results in Figure 5(f) show that the application of DUHL is not limited to sparse problems and SVMs. Even for ridge regression DUHL significantly outperforms all the reference scheme considered in this study.

## 6  Conclusion

We have presented a novel theoretical analysis of block coordinate descent, highlighting how the performance depends on the coordinate selection. These results prove that the contribution of individual coordinates to the overall duality gap is indicative of their relevance to the overall model optimization. Using this measure we develop a generic scheme for efficient training in the presence of high performance resources of limited memory capacity. We propose DUHL, an efficient gap memory-based strategy to select which part of the data to make available for fast processing. On a large dataset which exceeds the capacity of a modern GPU, we demonstrate that our scheme outperforms existing sequential approaches by over $10\times$ for Lasso and SVM models. Our results show that the practical gain matches the improved convergence predicted by our theory for gap-based sampling under the given memory and communication constraints, highlighting the versatility of the approach.

## Footnotes

[1]For $h : \mathbb{R}^d \to \mathbb{R}$ the convex conjugate is defined as $h^*(\mathbf{v}) := \sup_{\mathbf{u}\in\mathbb{R}^d} \mathbf{v}^\top\mathbf{u} - h(\mathbf{u})$.

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
