[Supplementary Material]

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

# Appendix

Organization of the appendix: We state detailed proofs of Theorem 1 and Theorem 2 in Appendix A. Then, we give some background information on coordinate descent and the local subproblem in Appendix B and C respectively. In Appendix D we then present details on the generalization of the TPA-SCD algorithm to SVM as well as Lasso. We provide exact expressions for the local updates, which together with the expression for the duality gap in Appendix E should guide the reader on how to easily practically implement our scheme for the different settings considered in the experiments.

## A  Proofs

In this section we state the detailed proofs of Theorem 1 and Theorem 2.

### A.1  Key Lemma

**Lemma 3.** *Consider problem formulation* (1). *Let $f$ be $L$-smooth. Further, let $g_i$ be $\mu$-strongly convex with convexity parameter $\mu \geq 0 \ \forall i \in [n]$. For the case $\mu = 0$ we need the additional assumption of $g_i$ having bounded support. Then, in any iteration $t$ of Algorithm 1 on* (1)*, we denote the updated coordinate block by $\mathcal{P}$ with $|\mathcal{P}| = m$ and define*

$$\rho_{t,\mathcal{P}} := \frac{\frac{1}{m}\sum_{j\in\mathcal{P}}\text{gap}_j(\alpha_j^{(t)})}{\frac{1}{n}\sum_{i=1}^n \text{gap}_i(\alpha_i^{(t)})} \tag{14}$$

*Then, for any $s \in [0,1]$, it holds that*

$$\mathbb{E}_{\mathcal{P}}\left[\mathcal{O}(\boldsymbol{\alpha}^{(t)}) - \mathcal{O}(\boldsymbol{\alpha}^{(t+1)})|\boldsymbol{\alpha}^{(t)}\right] \geq \theta\left[s\frac{m}{n}\mathbb{E}_{\mathcal{P}}\left[\rho_{t,\mathcal{P}}|\boldsymbol{\alpha}^{(t)}\right]\text{gap}(\boldsymbol{\alpha}^{(t)}) + \frac{s^2}{2}\gamma_{\mathcal{P}}^{(t)}\right] \tag{15}$$

*where*

$$\gamma_{\mathcal{P}}^{(t)} := \mathbb{E}_{\mathcal{P}}\left[\frac{\mu(1-s)}{s}\|\mathbf{u}^{(t)} - \boldsymbol{\alpha}^{(t)}\|^2 - L\|A(\mathbf{u}^{(t)} - \boldsymbol{\alpha}^{(t)})\|^2|\boldsymbol{\alpha}^{(t)}\right]. \tag{16}$$

*and $u_i^{(t)} \in \partial g_i^*(-\mathbf{a}_i^\top\mathbf{w}(\boldsymbol{\alpha}^{(t)}))$.*

*Proof.* First note that in every round of Algorithm 1, $\boldsymbol{\alpha}^{(t)} \to \boldsymbol{\alpha}^{(t+1)}$, only coordinates $i \in \mathcal{P}$ are changed and a $\theta$-approximate solution is computed on these coordinates. Hence, the improvement $\Delta_{\mathcal{O}}^t := \mathcal{O}(\boldsymbol{\alpha}^{(t)}) - \mathcal{O}(\boldsymbol{\alpha}^{(t+1)})$ in the objective (1) can be written as

$$\begin{aligned}
\Delta_{\mathcal{O}}^t &= \mathcal{O}(\boldsymbol{\alpha}^{(t)}) - \mathcal{O}(\boldsymbol{\alpha}^{(t)} + \Delta\boldsymbol{\alpha}_{[\mathcal{P}]}) \\
&\geq \mathcal{O}(\boldsymbol{\alpha}^{(t)}) - \left[(1-\theta)\mathcal{O}(\boldsymbol{\alpha}^{(t)}) + \theta\mathcal{O}(\boldsymbol{\alpha}^{(t)} + \Delta\boldsymbol{\alpha}_{[\mathcal{P}]}^\star)\right] \\
&= \theta\left[\mathcal{O}(\boldsymbol{\alpha}^{(t)}) - \min_{\Delta\boldsymbol{\alpha}_{[\mathcal{P}]}}\mathcal{O}(\boldsymbol{\alpha}^{(t)} + \Delta\boldsymbol{\alpha}_{[\mathcal{P}]})\right]. 
\end{aligned} \tag{17}$$

In order to lower bound (17) we look at a specific update direction: $\Delta\boldsymbol{\alpha}_{[\mathcal{P}]} = s(\mathbf{u}^{(t)} - \boldsymbol{\alpha}^{(t)})$ with $u_i^{(t)} \in \partial g_i^*(-\mathbf{a}_i^\top\mathbf{w}(\boldsymbol{\alpha}^{(t)}))$ for $i \in \mathcal{P}$ ($u_i^{(t)} = \alpha_i^{(t)}$ otherwise) and some $s \in [0,1]$. Note that for the subgradient to be well defined even for non-strongly convex functions $g_i$ we need the bounded support assumption on $g_i$.
This yields

$$\begin{aligned}
\Delta_{\mathcal{O}}^t &\geq \theta\left[\mathcal{O}(\boldsymbol{\alpha}^{(t)}) - \mathcal{O}(\boldsymbol{\alpha}^{(t)} + s(\mathbf{u}^{(t)} - \boldsymbol{\alpha}^{(t)}))\right] \\
&= \theta\big[\underbrace{f(A\boldsymbol{\alpha}^{(t)}) - f(A(\boldsymbol{\alpha}^{(t)} + s(\mathbf{u}^{(t)} - \boldsymbol{\alpha}^{(t)})))}_{\Delta f}\big] \\
&\qquad\qquad +\theta\sum_{i\in\mathcal{P}}\big[\underbrace{g_i(\alpha_i^{(t)}) - g_i(\alpha_i^{(t)} + s(u_i^{(t)} - \alpha_i^{(t)}))}_{\Delta_{g_i}}\big].
\end{aligned}$$

First, to bound $\Delta_f$ we use the fact that the function $f : \mathbb{R}^d \to \mathbb{R}$ has Lipschitz continuous gradient with constant $L$ which yields

$$
\begin{aligned}
\Delta_f &\geq -\left\langle \nabla f(A\boldsymbol{\alpha}^{(t)}), As(\mathbf{u}^{(t)} - \boldsymbol{\alpha}^{(t)}) \right\rangle - \frac{L}{2}\|As(\mathbf{u}^{(t)} - \boldsymbol{\alpha}^{(t)})\|^2 \\
&= -\sum_{i \in \mathcal{P}} \mathbf{a}_i^\top \mathbf{w}^{(t)} s(u_i^{(t)} - \alpha_i^{(t)}) - \frac{Ls^2}{2}\|A(\mathbf{u}^{(t)} - \boldsymbol{\alpha}^{(t)})\|^2.
\end{aligned}
\tag{18}
$$

Then, to bound $\Delta_{g_i}$ we use $\mu$-strong convexity of $g_i$ together with the Fenchel-Young inequality $g_i(u_i) \geq -u_i\mathbf{a}_i^\top\mathbf{w} - g_i^*(-\mathbf{a}_i^\top\mathbf{w})$ which holds with equality at $u_i \in \partial g_i^*(-\mathbf{a}_i^\top\mathbf{w})$ and find

$$
\begin{aligned}
\Delta_{g_i} &\geq -sg_i(u_i^{(t)}) + sg_i(\alpha_i^{(t)}) + \frac{\mu}{2}s(1-s)(u_i^{(t)} - \alpha_i^{(t)})^2 \\
&= su_i\mathbf{a}_i^\top\mathbf{w}^{(t)} + sg_i^*(-\mathbf{a}_i^\top\mathbf{w}^{(t)}) + sg_i(\alpha_i^{(t)}) + \frac{\mu}{2}s(1-s)(u_i^{(t)} - \alpha_i^{(t)})^2.
\end{aligned}
\tag{19}
$$

Finally, recalling the definition of the duality gap (4) and combining (18) and (19) yields

$$
\begin{aligned}
\Delta_{\mathcal{O}}^t &\geq \theta\,\Delta_f + \theta\sum_{i \in \mathcal{P}}\Delta_{g_i} \\
&\geq \theta\sum_{i \in \mathcal{P}} s\,\mathrm{gap}_i(\alpha_i^{(t)}) + \frac{\theta s^2}{2}\left[\frac{\mu(1-s)}{s}\|\mathbf{u}^{(t)} - \boldsymbol{\alpha}^{(t)}\|^2 - L\|A(\mathbf{u}^{(t)} - \boldsymbol{\alpha}^{(t)})\|^2\right].
\end{aligned}
$$

To conclude the proof we recall the definition of $\rho_{t,\mathcal{P}}$ in (7) and take the expectation over the choice of the coordinate block $\mathcal{P}$ which yields

$$
\mathbb{E}_{\mathcal{P}}\left[\mathcal{O}(\boldsymbol{\alpha}^{(t)}) - \mathcal{O}(\boldsymbol{\alpha}^{(t+1)})|\boldsymbol{\alpha}^{(t)}\right] \geq \theta s\frac{m}{n}\mathbb{E}_{\mathcal{P}}\left[\rho_{t,\mathcal{P}}|\boldsymbol{\alpha}^{(t)}\right]\mathrm{gap}(\boldsymbol{\alpha}^{(t)}) + \frac{\theta s^2}{2}\gamma_{\mathcal{P}}^{(t)}
\tag{20}
$$

with

$$
\gamma_{\mathcal{P}}^{(t)} := \mathbb{E}_{\mathcal{P}}\left[\frac{\mu(1-s)}{s}\|\mathbf{u}^{(t)} - \boldsymbol{\alpha}^{(t)}\|^2 - L\|A(\mathbf{u}^{(t)} - \boldsymbol{\alpha}^{(t)})\|^2\Big|\boldsymbol{\alpha}^{(t)}\right].
\tag{21}
$$

$\square$

## A.2 Proof Theorem 1

*Proof.* For strongly convex function $g_i$ we have $\mu > 0$ in Lemma 3. This allows us to choose $s$ such that $\gamma_{\mathcal{P}}^{(t)}$ in (15) vanishes. That is $s = \frac{\mu}{\frac{\sigma}{\beta}+\mu}$, where

$$
\sigma := \|A_{[\mathcal{P}]}\|^2 = \max_{\mathbf{v} \in \mathbb{R}^n}\frac{\|A_{[\mathcal{P}]}\mathbf{v}\|^2}{\|\mathbf{v}\|^2}.
\tag{22}
$$

This yields

$$
\mathbb{E}_{\mathcal{P}}\left[\mathcal{O}(\boldsymbol{\alpha}^{(t)}) - \mathcal{O}(\boldsymbol{\alpha}^{(t+1)})|\boldsymbol{\alpha}^{(t)}\right] \geq \theta s\frac{m}{n}\mathbb{E}_{\mathcal{P}}\left[\rho_{t,\mathcal{P}}|\boldsymbol{\alpha}^{(t)}\right]\mathrm{gap}(\boldsymbol{\alpha}^{(t)}).
$$

Now rearranging terms and exploiting that the duality gap always upper bounds the suboptimality we get the following recursion on the suboptimality $\epsilon^{(t)} := \mathcal{O}(\boldsymbol{\alpha}^{(t)}) - \mathcal{O}(\boldsymbol{\alpha}^\star)$:

$$
\mathbb{E}_{\mathcal{P}}\left[\epsilon^{(t+1)}|\boldsymbol{\alpha}^{(t)}\right] \leq \left(1 - \theta s\frac{m}{n}\mathbb{E}_{\mathcal{P}}\left[\rho_{t,\mathcal{P}}|\boldsymbol{\alpha}^{(t)}\right]\right)\epsilon^{(t)}.
$$

Defining $\eta_{\mathcal{P}} := \min_t \theta\mathbb{E}_{\mathcal{P}}\left[\rho_{t,\mathcal{P}} \mid \boldsymbol{\alpha}^{(t)}\right]$ and recursively applying the *tower property* of conditional expectations [15] which states

$$
\mathbb{E}_{\mathcal{P}}\left[\mathbb{E}_{\mathcal{P}}\left[\epsilon^{(t+1)}|\boldsymbol{\alpha}^{(t)}\right]|\boldsymbol{\alpha}^{(t-1)}\right] = \mathbb{E}_{\mathcal{P}}\left[\epsilon^{(t+1)}|\boldsymbol{\alpha}^{(t-1)}\right]
$$

we find

$$
\mathbb{E}_{\mathcal{P}}\left[\epsilon^{(t+1)}|\boldsymbol{\alpha}^{(0)}\right] \leq \left(1 - s\frac{m}{n}\eta_{\mathcal{P}}\right)^t\epsilon^{(0)}
$$

which concludes the proof. $\square$

### A.3 Proof Theorem 2

*Proof.* For the case where $\mu = 0$ Lemma 3 states:

$$\mathbb{E}_{\mathcal{P}}\left[\mathcal{O}(\boldsymbol{\alpha}^{(t)}) - \mathcal{O}(\boldsymbol{\alpha}^{(t+1)}) \mid \boldsymbol{\alpha}^{(t)}\right] \geq s\theta \frac{m}{n} \mathbb{E}_{\mathcal{P}}\left[\rho_{t,\mathcal{P}} | \boldsymbol{\alpha}^{(t)}\right] \mathrm{gap}(\boldsymbol{\alpha}^{(t)})$$
$$- \frac{\theta s^2}{2} L \mathbb{E}_{\mathcal{P}}\left[\|A(\mathbf{u}^{(t)} - \boldsymbol{\alpha}^{(t)})\|^2 \mid \boldsymbol{\alpha}^{(t)}\right].$$

Now rearranging terms, using $\sigma$ as defined in (22) and $\epsilon^{(t)} \leq \mathrm{gap}(\boldsymbol{\alpha}^{(t)})$, we find

$$\mathbb{E}_{\mathcal{P}}\left[\epsilon^{(t+1)} \mid \boldsymbol{\alpha}^{(t)}\right] \leq \left(1 - s\theta \frac{m}{n} \mathbb{E}_{\mathcal{P}}\left[\rho_{t,\mathcal{P}} \mid \boldsymbol{\alpha}^{(t)}\right]\right)\epsilon^{(t)} + \frac{\theta s^2}{2} L\sigma \mathbb{E}_{\mathcal{P}}\left[\|\mathbf{u}^{(t)} - \boldsymbol{\alpha}^{(t)}\|^2 \mid \boldsymbol{\alpha}^{(t)}\right].$$

In order to bound the last term in the above expression we use 1) the fact that $\sum_{i\in\mathcal{P}} g_i$ has $B$-bounded support which implies $\|\boldsymbol{\alpha}\| \leq B$ and 2) the duality between bounded support and Lipschitzness which implies $\|\mathbf{u}\| \leq B$ since $\mathbf{u} \in \partial \sum_{i\in\mathcal{P}} g_i^*(-\mathbf{a}_i^\top \mathbf{w})$. Then, by triangle inequality we find $\|\mathbf{u} - \boldsymbol{\alpha}\|^2 \leq 2B^2$ which yields the following recursion on the suboptimality for non strongly-convex $g_i$:

$$\mathbb{E}_{\mathcal{P}}\left[\epsilon^{(t+1)} \mid \boldsymbol{\alpha}^{(t)}\right] \leq \left(1 - s\theta \mathbb{E}_{\mathcal{P}}\left[\rho_{t,\mathcal{P}} \mid \boldsymbol{\alpha}^{(t)}\right] \frac{m}{n}\right)\epsilon^{(t)} + \frac{s^2}{2}\theta\gamma, \tag{23}$$

where $\gamma := 2LB^2\sigma$. Now defining $\eta_{\mathcal{P}} := \min_t \theta\, \mathbb{E}_{\mathcal{P}}\left[\rho_{t,\mathcal{P}} \mid \boldsymbol{\alpha}^{(t)}\right]$ and assuming $\eta_{\mathcal{P}} \geq 1$ , $\forall t$ we can upperbound the suboptimality at iteration $t$ as

$$\mathbb{E}_{\mathcal{P}}\left[\epsilon^{(t)} \mid \boldsymbol{\alpha}^{(0)}\right] \leq \frac{1}{\eta_{\mathcal{P}}m} \frac{2\gamma n^2}{2n + t - t_0} \tag{24}$$

with $t \geq t_0 = \max\left\{0, \frac{n}{m}\log\left(\frac{2\eta_{\mathcal{P}}m\epsilon^{(0)}}{\gamma n}\right)\right\}$.

Similar to [4] we prove this by induction:

$\underline{t = t_0}$: Choose $s := \frac{1}{\eta_{\mathcal{P}}}$ where $\eta_{\mathcal{P}} = \min_t \theta\mathbb{E}_{\mathcal{P}}\left[\rho_{t,\mathcal{P}} \mid \boldsymbol{\alpha}^{(t)}\right]$. Then at $t = t_0$, we have

$$\mathbb{E}_{\mathcal{P}}\left[\epsilon^{(t)} \mid \boldsymbol{\alpha}^{(0)}\right] \leq \left(1 - \frac{m}{n}\right)\mathbb{E}_{\mathcal{P}}\left[\epsilon^{(t-1)} \mid \boldsymbol{\alpha}^{(0)}\right] + \frac{s^2}{2}\theta\gamma$$

$$\leq \left(1 - \frac{m}{n}\right)^t \epsilon^{(0)} + \sum_{i=0}^{t-1}\left(1 - \frac{m}{n}\right)^i \frac{\theta\gamma}{2\eta^2}$$

$$\leq \left(1 - \frac{m}{n}\right)^t \epsilon^{(0)} + \frac{1}{1 - (1 - m/n)}\frac{\theta\gamma}{2\eta^2}$$

$$\leq e^{-tm/n}\epsilon^{(0)} + \frac{\theta n\gamma}{2m\eta_{\mathcal{P}}^2}$$

$$\overset{\theta < \eta}{\leq} \frac{n\gamma}{m\eta_{\mathcal{P}}}.$$

$\underline{t > t_0}$: For $t > t_0$ we use an inductive argument. Suppose the claim holds for $t$, giving

$$\mathbb{E}_{\mathcal{P}}\left[\epsilon^{(t)} \mid \boldsymbol{\alpha}^{(t-1)}\right] \leq \left(1 - \theta\mathbb{E}_{\mathcal{P}}\left[\rho_{t-1,\mathcal{P}} \mid \boldsymbol{\alpha}^{(t-1)}\right]\frac{s\,m}{n}\right)\epsilon^{(t-1)} - \frac{s^2}{2}\frac{m}{n}\theta\gamma,$$

$$\leq \left(1 - \eta_{\mathcal{P}}\frac{s\,m}{n}\right)\frac{1}{\eta_{\mathcal{P}}}\frac{2\gamma n}{2n + (t-1) - t_0} - \frac{s^2}{2}\frac{m}{n}\theta\gamma,$$

then, choosing $s = \frac{2n}{2n+(t-1)-t_0} \in [0,1]$ and applying the tower property of conditional expectations we find

$$
\begin{aligned}
\mathbb{E}_{\mathcal{P}}\left[\epsilon^{(t)} \mid \boldsymbol{\alpha}^{(0)}\right] \quad &\leq \quad \left(1 - \frac{2m\eta_{\mathcal{P}}}{2n+(t-1)-t_0}\right)\frac{1}{\eta_{\mathcal{P}}}\frac{2\gamma n}{2n+(t-1)-t_0} \\
&\quad + \left(\frac{2n}{2n+(t-1)-t_0}\right)^2 \frac{m}{n}\frac{\theta\gamma}{2} \\
&\overset{\theta \leq 1}{\leq} \quad \left(1 - \frac{m\eta_{\mathcal{P}}}{2n+(t-1)-t_0}\right)\frac{1}{\eta_{\mathcal{P}}}\frac{2\gamma n}{2n+(t-1)-t_0} \\
&= \quad \frac{1}{\eta_{\mathcal{P}}}\frac{2\gamma n}{(2n+(t-1)-t_0)}\frac{2n+(t-1)-t_0-m\eta_{\mathcal{P}}}{2n+(t-1)-t_0} \\
&\leq \quad \frac{1}{\eta_{\mathcal{P}}}\frac{2\gamma n}{(2n+t-t_0)}.
\end{aligned}
$$

$\square$

## B  Coordinate Descent

The classical coordinate descent scheme as described in Algorithm 3 solves for a single coordinate exactly in every round. This algorithm can be recovered as a special case of approximate block coordinate descent presented in Algorithm 1 where $m = 1$ and $\theta = 1$. In this case, similar to $\rho_{t,\mathcal{P}}$ we define

$$
\rho_{t,i} := \frac{\mathrm{gap}_i(\alpha_i^{(t)})}{\frac{1}{n}\sum_{j\in[n]}\mathrm{gap}_j(\alpha_j^{(t)})} \tag{25}
$$

which quantifies how much a single coordinate $i$ of iterate $\boldsymbol{\alpha}^{(t)}$ contributes to the duality gap (4).

**Strongly-convex** $g_i$**.**  Using Theorem 1 we find that for Algorithm 3 running on (1) where $f$ is $L$-smooth and $g_i$ is $\mu$-strongly convex with $\mu > 0$ for all $i \in [n]$, it holds that

$$
\mathbb{E}_j[\epsilon^{(t)} \mid \boldsymbol{\alpha}^{(0)}] \leq \left(1 - \rho_{\min}\left[\frac{\mu}{\mu+LR^2}\right]\frac{1}{n}\right)^t \epsilon^{(0)}, \tag{26}
$$

where $R$ upper bounds the column norm of $A$ as $\|\mathbf{a}_i\| \leq R \; \forall i \in [n]$, $\rho_{\min} := \min_t \mathbb{E}_j[\rho_{t,j} \mid \boldsymbol{\alpha}^{(t)}]$ and expectations are taken over the sampling distribution.

**General convex** $g_i$**.**  Using Theorem 2 we find that for Algorithm 3 running on (1) where $f$ is $L$-smooth and $g_i$ has $B$-bounded support for all $i \in [n]$ it holds that

$$
\mathbb{E}_j[\epsilon^{(t)} \mid \boldsymbol{\alpha}^{(0)}] \leq \frac{1}{\rho_{\min}}\frac{2\gamma n^2}{2n+t-t_0} \tag{27}
$$

with $t \geq t_0 = \max\left\{0, n\log\left(\frac{2\rho_{\min}\epsilon^{(0)}}{\gamma n}\right)\right\}$ and $\gamma = 2LB^2R^2$.

Note that these two results also cover widely used uniform sampling as a special case, where the coordinate $j$ in step 3 of Algorithm 3 is sampled uniformly at random and hence $\mathbb{E}_j\left[\rho_{t,j} \mid \boldsymbol{\alpha}^{(t)}\right] = 1$ which yields $\rho_{\min} = 1$. In this case we exactly recover the convergence results of [4, 13].

---

**Algorithm 3** Coordinate Descent

---

1: Initialize $\boldsymbol{\alpha}^{(0)} = \mathbf{0}$
2: **for** $t = 0, 1, 2, .....$ **do**
3:   select coordinate $i$
4:   $\Delta\alpha_i = \arg\min_{\Delta\alpha} \mathcal{O}(\boldsymbol{\alpha} + \mathbf{e}_i\Delta\alpha)$
5:   $\boldsymbol{\alpha}^{(t+1)} = \boldsymbol{\alpha}^{(t)} + \mathbf{e}_i\Delta\alpha_i$
6: **end for**

---

## C  Local Subproblem

In Section 4.1, we have suggested to replace the local optimization problem in Step 4 of Algorithm 1 with a simpler quadratic local problem. More precisely, to replace

$$\underset{\Delta\boldsymbol{\alpha}_{[\mathcal{P}]}\in\mathbb{R}^n}{\arg\min}\ f(A(\boldsymbol{\alpha}+\Delta\boldsymbol{\alpha}_{[\mathcal{P}]})) + \sum_{i\in\mathcal{P}} g_i((\boldsymbol{\alpha}+\Delta\boldsymbol{\alpha})_i) \tag{5}$$

by instead

$$\underset{\Delta\boldsymbol{\alpha}_{[\mathcal{P}]}\in\mathbb{R}^n}{\arg\min}\ f(A\boldsymbol{\alpha}) + \nabla f(A\boldsymbol{\alpha})^\top A\Delta\boldsymbol{\alpha}_{[\mathcal{P}]} + \frac{L}{2}\|A\Delta\boldsymbol{\alpha}_{[\mathcal{P}]}\|_2^2 + \sum_{i\in\mathcal{P}} g_i((\boldsymbol{\alpha}+\Delta\boldsymbol{\alpha})_i). \tag{12}$$

Note that the modified objective (12) does not depend on $\mathbf{a}_i$ for $i\notin\mathcal{P}$ other than through $\mathbf{v}$. Thus, (12) can be solved locally on processing unit $\mathcal{B}$ with only access to $A_{[\mathcal{P}]}$ (columns $\mathbf{a}_i$ of A with $i\in\mathcal{P}$) and the current shared state $\mathbf{v} := A\boldsymbol{\alpha}$. Note that for quadratic functions $f$ the two problems (5) and (12) are equivalent. This applies to ridge regression, Lasso as well as $L_2$-regularized SVM.

For functions $f$ where the Hessian $\nabla^2 f$ cannot be expressed as a scaled identity, (12) forms a second-order upper-bound on the objective (5) by $L$-smoothness of $f$.

**Proposition 4.** *The convergence results of Theorem 1 and 2 similarly hold if the update in Step 4 of Algorithm 1 is performed on* (12) *instead of* (5)*, i.e., a $\theta$-approximate solution is computed on the modified objective* (12)*.*

*Proof.* Let us define

$$\tilde{\mathcal{O}}(\boldsymbol{\alpha}^{(t)},\mathbf{v},\Delta\boldsymbol{\alpha}_{[\mathcal{P}]}) \ \ := \ \ f(A\boldsymbol{\alpha}) + \nabla f(A\boldsymbol{\alpha})^\top A\Delta\boldsymbol{\alpha}_{[\mathcal{P}]} + \frac{L}{2}\|A\Delta\boldsymbol{\alpha}_{[\mathcal{P}]}\|_2^2 + \sum_{i\in\mathcal{P}} g_i((\boldsymbol{\alpha}+\Delta\boldsymbol{\alpha})_i)$$

Assume the update step $\Delta\boldsymbol{\alpha}_{[\mathcal{P}]}$ performed in Step 4 of Algorithm 1 is a $\theta$-approximate solution to (12), then we can bound the per-step improvement in any iteration $t$ as:

$$\begin{aligned}
\mathcal{O}(\boldsymbol{\alpha}^{(t)}) - \mathcal{O}(\boldsymbol{\alpha}^{(t+1)}) \ &\geq\ \mathcal{O}(\boldsymbol{\alpha}^{(t)}) - \tilde{\mathcal{O}}(\boldsymbol{\alpha}^{(t)},\mathbf{v},\Delta\boldsymbol{\alpha}_{[\mathcal{P}]}) \\
&\geq\ \mathcal{O}(\boldsymbol{\alpha}^{(t)}) - \Big[\theta\min_{\mathbf{s}_{[\mathcal{P}]}}\tilde{\mathcal{O}}(\boldsymbol{\alpha}^{(t)},\mathbf{v},\mathbf{s}_{[\mathcal{P}]}) + (1-\theta)\tilde{\mathcal{O}}(\boldsymbol{\alpha}^{(t)},\mathbf{v},\mathbf{0})\Big] \\
&=\ \theta\Big[\mathcal{O}(\boldsymbol{\alpha}^{(t)}) - \min_{\mathbf{s}_{[\mathcal{P}]}}\tilde{\mathcal{O}}(\boldsymbol{\alpha}^{(t)},\mathbf{v},\mathbf{s}_{[\mathcal{P}]})\Big].
\end{aligned}$$

where we used $\tilde{\mathcal{O}}(\boldsymbol{\alpha}^{(t)},\mathbf{v},\mathbf{0}) = \mathcal{O}(\boldsymbol{\alpha}^{(t)})$ and $\mathcal{O}(\boldsymbol{\alpha}^{(t)}+\Delta\boldsymbol{\alpha}_{[\mathcal{P}]}) \leq \tilde{\mathcal{O}}(\boldsymbol{\alpha}^{(t)},\mathbf{v},\Delta\boldsymbol{\alpha}_{[\mathcal{P}]})$ which follows by smoothness of $f$. Hence, the following inequality holds for an arbitrary block update $\tilde{\mathbf{s}}_{[\mathcal{P}]}$:

$$\mathcal{O}(\boldsymbol{\alpha}^{(t)}) - \mathcal{O}(\boldsymbol{\alpha}^{(t+1)}) \geq \theta\ \Big[\mathcal{O}(\boldsymbol{\alpha}^{(t)}) - \tilde{\mathcal{O}}(\boldsymbol{\alpha}^{(t)},\mathbf{v},\tilde{\mathbf{s}}_{[\mathcal{P}]})\Big] \tag{28}$$

Now, if we plug in the definitions of $\mathcal{O}(\boldsymbol{\alpha}^{(t)})$ and $\tilde{\mathcal{O}}(\boldsymbol{\alpha}^{(t)},\mathbf{v},\tilde{\mathbf{s}}_{\mathcal{P}})$, then split the expression into terms involving $f$ and terms involving $g_i$ as in Section A.1 and consider the same specific update direction, (i.e. $\tilde{\mathbf{s}} = s(\mathbf{u}-\boldsymbol{\alpha})$ where $u_i \in g_i^*(-\mathbf{a}_i^\top\mathbf{w})$, $s\in[0,1]$), we recover the bounds (19) and (18) for the respective terms. If we then proceed along the lines of Section A we get exactly the same bound on the per step improvement as in (15). The convergence guarantees from Theorem 1 and Theorem 2 follow immediately. $\qquad\square$

### C.1 Examples

For completeness, we state the local subproblem formulation explicitly for the objectives considered in the experiments.

**a) Ridge regression.**  The ridge regression objective is given by

$$\min_{\boldsymbol{\alpha}\in\mathbb{R}^n} \quad \frac{1}{2d}\|A\boldsymbol{\alpha}-\mathbf{b}\|_2^2 + \frac{\lambda}{2}\|\boldsymbol{\alpha}\|_2^2, \tag{29}$$

where $\mathbf{b}\in\mathbb{R}^d$ denotes the vector of labels. For (29) the local subproblem (12) can be stated as

$$\underset{\Delta\boldsymbol{\alpha}_{[\mathcal{P}]}\in\mathbb{R}^n}{\arg\min} \quad \frac{1}{2d}\Big\|\sum_{i\in\mathcal{P}}\mathbf{a}_i\Delta\boldsymbol{\alpha}_{[\mathcal{P}]i}\Big\|_2^2 + \frac{1}{d}\sum_{i\in\mathcal{P}}(\mathbf{v}-\mathbf{b})^\top\mathbf{a}_i\Delta\boldsymbol{\alpha}_{[\mathcal{P}]i} + \frac{\lambda}{2}\sum_{i\in\mathcal{P}}(\boldsymbol{\alpha}+\Delta\boldsymbol{\alpha}_{[\mathcal{P}]})_i^2.$$

**b) Lasso.**  For the Lasso objective

$$\min_{\boldsymbol{\alpha}\in\mathbb{R}^n} \quad \frac{1}{2d}\|A\boldsymbol{\alpha}-\mathbf{b}\|_2^2 + \lambda\|\boldsymbol{\alpha}\|_1, \tag{30}$$

where $\mathbf{b}\in\mathbb{R}^d$ denotes the vector of labels, the local problem (12) can similarly be stated as

$$\underset{\Delta\boldsymbol{\alpha}_{[\mathcal{P}]}\in\mathbb{R}^n}{\arg\min} \quad \frac{1}{2d}\Big\|\sum_{i\in\mathcal{P}}\mathbf{a}_i\Delta\boldsymbol{\alpha}_{[\mathcal{P}]i}\Big\|_2^2 + \frac{1}{d}\sum_{i\in\mathcal{P}}(\mathbf{v}-\mathbf{b})^\top\mathbf{a}_i\Delta\boldsymbol{\alpha}_{[\mathcal{P}]i} + \lambda\sum_{i\in\mathcal{P}}|(\boldsymbol{\alpha}+\Delta\boldsymbol{\alpha}_{[\mathcal{P}]})_i|.$$

**c) $L_2$-regularized SVM.**  In case of the $L_2$-regularized SVM problem we consider the dual problem formulation

$$\min_{\boldsymbol{\alpha}\in\mathbb{R}^n} \quad \frac{1}{n}\sum_i(-y_i\alpha_i) + \frac{1}{2\lambda n^2}\|A\boldsymbol{\alpha}\|_2^2, \tag{31}$$

with $y_i\alpha_i\in[0,1]$, $\forall i$, where column $\mathbf{a}_i$ of $A$ corresponds to sample $i$ with corresponding label $y_i$. The local subproblem (12) for (31) can then be stated as

$$\underset{\Delta\boldsymbol{\alpha}_{[\mathcal{P}]}\in\mathbb{R}^n}{\arg\min} \quad \frac{1}{n}\sum_{i\in\mathcal{P}}(-y_i(\boldsymbol{\alpha}+\Delta\boldsymbol{\alpha}_{[\mathcal{P}]})_i) + \frac{1}{2\lambda n^2}\Big\|\sum_{i\in\mathcal{P}}\mathbf{a}_i\Delta\boldsymbol{\alpha}_{[\mathcal{P}]i}\Big\|_2^2 + \frac{1}{\lambda n^2}\sum_{i\in\mathcal{P}}\mathbf{v}^\top\mathbf{a}_i\Delta\boldsymbol{\alpha}_{[\mathcal{P}]i}$$

subject to $y_i(\boldsymbol{\alpha}+\Delta\boldsymbol{\alpha}_{[\mathcal{P}]})_i\in[0,1]$ for $i\in\mathcal{P}$.

## D  Generalization of TPA-SCD

TPA-SCD is presented in [10] as an efficient GPU solver for the ridge regression problem. TPA-SCD implements an asynchronous version of stochastic coordinate descent especially suited for the GPU architecture. Every coordinate is updated by a dedicated thread block and these thread blocks are scheduled for execution in parallel on the available streaming multiprocessors of the GPU. Individual coordinate updates are computed by solving for this coordinate exactly while keeping all the others fixed. To synchronize the work between threads, the vector $\tilde{\mathbf{v}} := A\boldsymbol{\alpha}-\mathbf{b}$ is written to the GPU main memory and shared among all threads. To keep $\boldsymbol{\alpha}$ and $\tilde{\mathbf{v}}$ consistent $\tilde{\mathbf{v}}$ is updated asynchronously by the thread blocks after every single coordinate update to $\boldsymbol{\alpha}$ exploiting the atomic add operation of modern GPUs.

## D.1 Elastic Net

The generalization of the TPA-SCD algorithm from $L_2$ regularization to elastic net regularized problems including Lasso is straightforward. Let us consider the following objective:

$$\min_{\boldsymbol{\alpha} \in \mathbb{R}^n} \frac{1}{2d} \|A\boldsymbol{\alpha} - \mathbf{b}\|_2^2 + \lambda \left( \frac{\eta}{2} \|\boldsymbol{\alpha}\|_2^2 + (1-\eta)\|\boldsymbol{\alpha}\|_1 \right) \tag{32}$$

with trade-off parameter $\eta \in [0, 1]$.

In this case the only difference to the ridge regression solver presented in [10] is the computation of the individual coordinate updates in [10, Algorithm 2]. That is, solving for a single coordinate $j$ exactly in (32) yields the following update rule:

$$\alpha_j^{t+1} = \text{sign}(\gamma) \left[ |\gamma| - \tau \right]_+ \tag{33}$$

with soft-thresholding parameter

$$\tau = \frac{\lambda d(1-\eta)}{\|\mathbf{a}_j\|_2^2 + \lambda \eta d} \tag{34}$$

and

$$\gamma = \frac{\alpha_j^t \|\mathbf{a}_j\|_2^2 - \mathbf{a}_j^\top \tilde{\mathbf{v}}^t}{\|\mathbf{a}_j\|_2^2 + \lambda \eta d}. \tag{35}$$

Here $\tilde{\mathbf{v}}^t$ denotes the current state of the shared vector $\tilde{\mathbf{v}}^t := A\boldsymbol{\alpha}^t - \mathbf{b}$ which is updated after every coordinate update as

$$\tilde{\mathbf{v}}^{t+1} = \tilde{\mathbf{v}}^t + \mathbf{a}_j(\alpha_j^{t+1} - \alpha_j^t).$$

Similar to ridge regression we parallelize the computation of $\mathbf{a}_j^\top \tilde{\mathbf{v}}^t$ and $\mathbf{a}_j^\top \mathbf{a}_j$ in (34) and (35) in every iteration over all threads of the thread block in order to fully exploit the parallelism of the GPU.

## D.2 $L_2$-regularized SVM

TPA-SCD can also be generalized to optimize the dual SVM objective (31). In the dual formulation (31) a block of coordinates $\mathcal{P}$ of $\boldsymbol{\alpha}$ corresponds to a subset of samples (as opposed to features). Hence, individual thread blocks in TPA-SCD optimize for a single sample at a time where the share information corresponds to $\hat{\mathbf{v}} := A\boldsymbol{\alpha}$ (instead of $A\boldsymbol{\alpha} - \mathbf{b}$ as in the ridge regression implementation which only impacts initialization of the shared vector). The corresponding single coordinate update can then be computed as

$$\Delta\alpha_j = \frac{y_j - \frac{1}{\lambda n}\mathbf{a}_j^\top \hat{\mathbf{v}}^t}{\frac{1}{\lambda n}\|\mathbf{a}_j\|_2^2} \tag{36}$$

and incorporating the constraint $(y_i\alpha_i \in [0, 1], \forall i)$ we find:

$$\alpha_j^{t+1} = y_j \max(0, \min(1, y_j(\alpha_j^t + \Delta\alpha_j)))$$

and update $\hat{\mathbf{v}}$ accordingly:

$$\hat{\mathbf{v}}^{t+1} = \hat{\mathbf{v}}^t + \mathbf{a}_j(\alpha_j^{t+1} - \alpha_j^t).$$

Again, multiple threads in a thread block can be used to compute individual updates by parallelizing the computation of $\mathbf{a}_j^\top \mathbf{v}$ and $\mathbf{a}_j^\top \mathbf{a}_j$ for every update.

# E Duality Gap

The computation of the duality gap is essential for the implementation of the selection scheme in Algorithm 2. We therefore devote this section to explicitly state the duality gap for the objective functions considered in our experiments.

**Ridge regression.** Since the $L_2$-norm is self-dual the computation of the duality gap for the ridge regression objective (29) is straightforward:

$$\mathrm{gap}(\boldsymbol{\alpha}) \;=\; \frac{1}{d} \left[ \sum_{i\in[n]} \alpha_i\, \mathbf{a}_i^\top \mathbf{w} + \frac{1}{2\lambda d}(\mathbf{a}_i^\top \mathbf{w})^2 + \lambda d\frac{1}{2}\alpha_i^2 \right]$$

where $\mathbf{w} := A\boldsymbol{\alpha} - \mathbf{b}$.

**Lasso.** In order to compute a valid duality gap for the Lasso problem (30) we need to employ the Lipschitzing trick as suggested in [4]. This enables to compute a globally defined duality gap even for non-bounded conjugate functions $g_i^*$ such as when the $g_i$ form the $L_1$ norm. The Lipschitzing trick is applied coordinate-wise to every $g_i := |\cdot|$. It artificially bounds the support of $g_i$, where we choose the bound $B$ such that $\|\boldsymbol{\alpha}^{(t)}\|_1 \leq B\ \forall t > 0$, and hence $|\alpha_i^t| \leq B, \forall i, t$. Thus every iterate $\boldsymbol{\alpha}^{(t)}$ is guaranteed to lie within the support. This choice further guarantees that the bounded support modification does not affect the optimization and the original objective is untouched inside the region of interest. For the Lasso objective (30) we can satisfy this with the following choice: $B = \frac{f(0)}{\lambda d} = \frac{\|\mathbf{b}\|_2^2}{2\lambda d}$. Given $B$, the duality gap for the Lasso problem can be computed as

$$\mathrm{gap}(\boldsymbol{\alpha}) \;=\; \frac{1}{d} \left[ \sum_{i\in[n]} \alpha_i\, \mathbf{a}_i^\top \mathbf{w} + B\left[|\mathbf{a}_i^\top \mathbf{w}| - \lambda d\right]_+ + \lambda d|\alpha_i| \right]$$

where we recall the primal-dual mapping:

$$\mathbf{w} := A\boldsymbol{\alpha} - \mathbf{b}.$$

**$L_2$-regularized SVM.** The $L_2$-regularized SVM objective is given as

$$\mathcal{P}(\mathbf{w}) = \frac{1}{n} \sum_{i\in[n]} h_i(\mathbf{a}_i^\top \mathbf{w}) + \frac{\lambda}{2}\|\mathbf{w}\|_2^2 \tag{37}$$

where for every $i \in [n]$, $h_i(u) = \max\{0, 1 - y_i u\}$ denotes the hinge loss and $\mathbf{a}_i$ sample $i$ with label $y_i$. The corresponding dual problem formulation is given in (31). The duality gap (2) for the $L_2$-regularized SVM objective can be computed as follows:

$$\mathrm{gap}(\boldsymbol{\alpha}) \;=\; \frac{1}{n} \left[ \sum_{i\in[n]} \alpha_i \mathbf{a}_i^\top \mathbf{w} + h_i(\mathbf{a}_i^\top \mathbf{w}) - y_i\alpha_i \right]$$

where the primal-dual mapping is given as

$$\mathbf{w} := \frac{1}{n\lambda} A\boldsymbol{\alpha}.$$