[Reviews · NeurIPS 2017]

Reviewer 1



This paper proposes a primal-dual block coordinate descent with active block selection based on coordinate-wise duality gaps for separable penalty function. The convergence results account for approximate solution of block update (used when particular machine doesn’t have access to complete data) and greedy block selection scheme to reduce the gap by maximum possible at the current iteration. The algorithm is shown to converge linearly, (c^t) in t iterations, for strongly convex penalty g and (c/t) in t iterations for bounded support penalty g. The algorithm is then applied to heterogeneous environment where machines with different memory and computation power are available. The numerical section compares the algorithm to other reference schemes like sequential or importance sampling based block selection and single threaded CPU. The paper is well written with clear organization and I enjoyed reading the proofs. I’m not an expert in this specific area and will leave the novelty judgment to other reviewers.

Reviewer 2



The paper presents a specific way of exploiting a system with two heterogeneous compute nodes with complementary capabilities. The idea is to perform block co-ordinate algorithm with the subproblem of optimizing with the block variables solved at the node with high-compute-low-memory and the subproblem of choosing the block at the node with low-compute-high-memory. Here are some comments: 1. Overall, it is not very clear why the above is the only worthwhile strategy to try. Atleast empirical comparison with popular distributed algorithms (that ignore the resource capabilities) is necessary in order to access the merit of this proposal. 2. More importantly, the proposed split-up of workload in inherently not parallelizable. Hence a time-delay strategy is employed. Though empirically it is shown that it works, it not clear why this is the way to go? 3. In literature of block co-ordinate descent scoring functions (similar to (6) ) are popular. An empirical comparison with strategies like those used in SMO (libsvm etc.; first order and second order criteria for sub-optimality) seems necessary. 4. The results of Theorem1 and Theorem2 are interesting (I have not been able to fully check the correctness of the proofs). 5. Refer eqn (2). The duality is lower-bounded by the expression on LHS in (2). It is need not be equal to it. I have read the rebuttal and would like to stay with the score.

Reviewer 3



This paper considers the use of compute accelerators (e.g., GPUs) to efficiently solve large-scale machine learning problems. The set-up used by the authors is that of a compute unit with a large memory but with small compute power, coupled with a low memory high compute unit. The paper also proposes a new algorithm called DUHL (Duality Gap-Based Heterogeneous Learning Scheme) which is a coordinate descent based algorithm that works well on the computational setup previously described. The theoretical results support the proposed approach, as do the numerical experiments. This is an interesting paper. However, there are a lot of ideas crammed into this work that I feel have not been explained fully (probably due to the space limitations). It feels like more could have been done to ensure that the paper is more cohesive, so it is easier for the reader to follow, and also more motivation for what is being proposed.